# Investigating p62 Concentrations in Cerebrospinal Fluid of Patients with Dementia: A Potential Autophagy Biomarker In Vivo?

**DOI:** 10.3390/brainsci12101414

**Published:** 2022-10-20

**Authors:** Elisa Rubino, Silvia Boschi, Fausto Roveta, Andrea Marcinnò, Aurora Cermelli, Cristina Borghese, Maria Claudia Vigliani, Innocenzo Rainero

**Affiliations:** 1Department of Neuroscience and Mental Health, AOU Città della Salute e della Scienza di Torino, 10126 Torino, Italy; 2Department of Neuroscience “Rita Levi Montalcini”, University of Torino, 10126 Torino, Italy

**Keywords:** *SQSTM1*/p62, biomarkers, autophagy, cerebrospinal fluid, Alzheimer’s disease, frontotemporal dementia

## Abstract

Several studies have revealed defects in autophagy in neurodegenerative disorders including Alzheimer’s disease (AD) and frontotemporal dementia (FTD). *SQSTM1*/p62 plays a key role in the autophagic machinery and may serve as a marker for autophagic flux in vivo. We investigated the role of p62 in neurodegeneration, analyzing its concentrations in the CSF of AD and FTD patients. We recruited 76 participants: 22 patients with AD, 28 patients with FTD, and 26 controls. CSF p62 concentrations were significantly increased in AD and FTD patients when compared to controls, which persisted after adjusting for age (*p* = 0.01 and *p* = 0.008, respectively). In female FTD patients, p62 positively correlated with the neurodegenerative biomarkers t-Tau and p-Tau. A significant correlation between CSF p62 concentrations and several clinical features of AD was found. Our data show that p62 is increased in CSF of AD and FTD patients, suggesting a key role of autophagy in these two disorders. The levels of p62 in CSF may reflect an altered autophagic flux, and p62 could represent a potential biomarker of neurodegeneration.

## 1. Introduction

Neurodegenerative disorders such as Alzheimer’s disease (AD), frontotemporal dementia (FTD), amyotrophic lateral sclerosis (ALS), and Parkinson’s disease (PD), are characterized by the progressive deposition of misfolded and aggregated proteins that lose their physiological functions and acquire neurotoxic effects [1]. In recent years, great scientific and economic efforts have been made towards the identification of more accurate and predictive biomarkers of neurodegeneration in vivo [2]. Although the underlying pathogenic mechanisms vary among different diseases, an alteration of autophagic flux is now considered a common mechanism of neurodegeneration [3].

Autophagy is characterized by a complex cellular process that leads to the clearance of misfolded or damaged proteins and dysfunctional organelles [4]. This complex machinery is induced by different signaling pathways such as oxidative stress and neuronal excitotoxicity. In healthy neurons, autophagy is observed at low basal levels [5] while activation of autophagy is well described in different neurodegenerative disorders [6]. Increasing evidence suggests that in both AD and FTD, there is an upregulation of autophagic machinery. Several studies have shown that neuronal autophagy is significantly impaired in AD [7,8]. In FTD, disease-causing genes such as *SQSTM1*, *OPTN*, and *VCP* are directly involved in protein degradation pathways as their products contribute to recruitment of ubiquitinated proteins to the autophagosome, a key component of autophagy [9].

Human p62 is a scaffold protein of 440 amino acids, coded by the *SQSTM1* gene and involved in several functions, including autophagy, apoptosis, inflammation, and cell survival, suggesting a role as a signaling hub that regulates cell viability in response to cytotoxic stress [10,11]. Mutations of *SQSTM1* gene were initially described in Paget’s disease of bone [12], a chronic progressive skeletal disorder; more recently, gene mutations were reported in ALS [13] and FTD patients [14,15]. In neuropathological findings, p62 is present in cytoplasmic aggregates in both ALS and FTD samples, and represents a common component of neurofibrillary tangles in AD [16].

Cerebrospinal fluid (CSF) is enriched in brain-derived substances, and CSF biomarkers might represent a valid instrument to assess neuropathology in vivo. At present, however, there are no established CSF biomarkers for monitoring brain autophagy in vivo. In 2017, Au et al. examined p62 concentrations in the CSF of 30 children with severe traumatic brain injury (TBI) and found that p62 concentrations were significantly increased, suggesting an impairment of the autophagic flux [17]. In addition, increased p62 concentrations were associated with an unfavorable outcome. More recently, several CSF autophagic proteins, including p62, were analyzed in a cohort of 32 patients with early PD, showing a dysregulation of the lysosomal autophagy pathway [18].

To date, a few studies have investigated CSF p62 concentrations across different neurodegenerative disorders. Thus, the primary aim of this study was to evaluate whether p62 concentrations were detectable in the cerebrospinal fluid of patients with AD or FTD, comparing them with a control population. In addition, we investigated correlations between this autophagy biomarker and clinical or radiological characteristics of AD and FTD.

## 2. Materials and Methods

Seventy-six consecutive patients referred to the “Rita Levi Montalcini” Department of Neuroscience of the University Hospital Città della Salute e della Scienza di Torino (Italy) were selected for the study. Twenty-two patients with AD (10 males and 12 females, mean age: 66.41 ± 8.01 years; age range: 47–80 years) and 28 patients with FTD (18 males and 10 females, mean age: 64 ± 7.98 years; age range: 50–78 years) were enrolled.

Clinical diagnosis of probable AD was established according to joint Alzheimer’s Association Workgroup and National Institute of Aging guidelines [19] and the ATN classification scheme [20]. The diagnosis for the behavioral variant of FTD was made according to the Rascovsky criteria [21], and the different forms of language variant were diagnosed according to specific established criteria [22]. Based on these criteria, 25 of the FTD patients were classified as behavioral variant FTD (bvFTD), 2 as nonfluent/agrammatic variant PPA (nfvPPA), and 1 as semantic variant primary progressive aphasia (svPPA). For each demented patient, neuroimaging findings (brain MRI, 18FDG-PET), neuropsychological evaluations, and CSF levels of AD core biomarkers Aβ42, total tau (t-Tau), and phosphorylated tau at threonine 181 (p-Tau) data were obtained. Patients with FTD were also sequenced for *GRN*, *MAPT*, *SQSTM1* (that encodes p62), and *C9orf72* genes.

Twenty-six cognitive-spared patients (10 males and 16 females, mean age: 56.18 ± 15.02 years; age range: 40–75 years) with neurological conditions other than neurodegenerative disorders were included as a control group, including patients with suspected acute polyneuropathy (n. 7), muscular dystrophy (n. 3), multiple sclerosis (n. 3), and normal pressure hydrocephalus (n. 13). The demographic and clinical features are summarized in Table 1. The local Ethics Committee approved the protocol of the study.

CSF sampling and collection were performed according to previously published protocols and guidelines [23]. After overnight fasting, CSF samples were collected in sterile polypropylene tubes via a lumbar puncture performed between 9 a.m. and 12 a.m. CSF samples were centrifuged at 1500× *g* for 10 min at room temperature, subsequently transferred into aliquot tubes, and stored in a −80 °C freezer.

The CSF p62 levels were evaluated using a commercially available enzyme-linked immunosorbent assay (ELISA) kit (Cloud-Clone Corp). The limit of detection was 0.1 ng/mL for p62, and samples below the level of detection were assigned a value of 0 ng/mL. All samples were assayed in duplicate. CSF concentrations of Aβ42, t-Tau, and p-Tau181 were also evaluated using the ELISA method with available commercially Innotest Kits (Fujirebio), according to the manufacturer’s instructions.

All statistical analyses were performed using SPSS version 20 (SPSS Inc, Chicago, IL, USA). Continuous variables were presented as means ± standard deviations or medians and range, and categorical ones as count and percentage. We compared diagnostic groups (AD, FTD and controls) using ANOVA for normally distributed variables and Kruskal–Wallis and Mann–Whitney U tests for non-normally distributed variables, as appropriate. Correlations between p62 and CSF biomarkers were also evaluated using Pearson’s test, also adjusted for age. The statistical models were adjusted for possible confounders. A two-tailed *p* value of <0.05 was considered significant in biochemical and clinical comparisons.

## 3. Results

### 3.1. Overall Population

No difference in education, comorbidities, or smoking habits was found between cases and controls. The gender distribution was comparable in the control and FTD/AD cohorts. Patients with AD and FTD did not differ in age, whereas control subjects were younger than patients with dementia (Table 1).

### 3.2. Analysis of p62 in AD and FTD Patients

The p62 concentrations were found to be significantly different between AD patients and controls (0.74 ± 0.64 ng/mL vs. 0.15 ± 0.33 ng/mL, *p* = 0.011) (Figure 1), which persisted after adjusting for age (*p* = 0.01). The p62 concentrations were not different in AD males with respect to male controls, whereas female AD patients showed higher p62 levels when compared to female controls (0.58 ± 0.54 ng/mL vs. 0.07 ± 0.29 ng/mL, *p* = 0.044) (Figure 2). No significant difference in CSF p62 concentration was found between FTD and AD patients (*p* = 0.779).

Crude p62 levels were significantly increased in FTD patients when compared to controls (0.97 ± 0.99 ng/mL vs. 0.15 ± 0.33 ng/mL, *p* < 0.001) (Figure 1), which persisted after adjusting for age (*p* = 0.008). When analyzing the gender effect, p62 concentrations were found to be significantly different between male FTD patients and male controls (1.17 ± 1.06 ng/mL vs. 0.20 ± 0.41 ng/mL, *p* = 0.029), and between female FTD patients and female controls (0.61 ± 0.76 ng/mL vs. 0.07 ± 0.29 ng/mL, *p* = 0.044).

### 3.3. Relationship between p62 Levels and AD CSF Core Biomarker Levels, Aβ42, t-Tau, and p-Tau

We hypothesized that the levels of p62 would also affect neurodegenerative processes. Therefore, we examined the relationship between p62 levels and CSF AD core biomarkers Aβ42, t-Tau, and p-Tau. CSF p62 levels did not correlate with biomarkers of neurodegeneration in the AD group, which persisted when adjusted for age. In contrast, CSF p62 concentrations correlated positively with the neurodegenerative biomarkers t-Tau (r = 0.853, *p* = 0.002) and p-Tau in the female FTD subgroup alone (r = 0.807, *p* = 0.005) (Figure 3).

### 3.4. P62 and Clinical Characteristics

Several clinical features emerged as significantly linked to CSF p62 concentrations in the AD group. In AD patients, p62 levels were negatively associated and with the presence of language disturbance (r = −0.438; *p* = 0.041) (Figure 4a). In AD males, p62 concentrations were inversely associated with the presence of temporal and spatial disorientation (r = −0.674; *p* = 0.033), and depression (r = −0.781; *p* = 0.008). Furthermore, p62 concentrations were negatively correlated with temporal atrophy in the AD subgroup (r = −0.529; *p* = 0.011), mainly in the male subgroup (r = −0.854; *p* = 0.002). No correlation between p62 levels and MMSE was found (Figure 4b). Finally, no correlation was found between p62 concentrations and the severity of disease.

No correlations between p62 levels and the collected clinical characteristics of FTD patients were found. In particular, no correlation between p62 CSF concentration and neuropsychological test results was found (including MMSE and Frontal Assessment Battery). In addition, we did not find any correlations between p62 levels and severity of the disease.

In this cohort of patients, we did not find any mutations in the *SQSTM1* gene; we only found the single-nucleotide polymorphism rs186996560 in the *SQSTM1* in a FTD patient. In our FTD group, p62 levels were elevated in patients both carrying and not carrying mutations in known causative genes in respect to controls. No significant correlation was found between p62 levels and the presence or absence of gene mutations (*p* = 0.317).

## 4. Discussion

To our knowledge, this is the first study to investigate CSF p62 concentrations as marker of autophagy in patients with dementia. We showed that CSF p62 concentrations were significantly increased both in AD and FTD patients in comparison to controls. Furthermore, in female patients with FTD, we found that CSF p62 concentrations correlated with markers of neurodegeneration like p-Tau and t-Tau. In patients with AD, several neuropsychological characteristics of the disease showed a significant correlation with the investigated biomarker of autophagy. Taken together, our data support the hypothesis that CSF p62 concentrations may be a useful in vivo biomarker of neuronal autophagy.

Increased CSF p62 concentrations in patients with neurodegenerative disorders may have several neurobiological explanations. First, p62 is involved in several cellular processes relevant to aging, including protein degradation [24], mitophagy [25], and inflammation. It also seems to be essential for aggregation and autophagic clearance of ubiquitinated proteins [26,27]. The increase in p62 in patients with neurodegenerative disorders could be the result of promoting autophagic clearance of protein aggregates (known as aggrephagy). Loss of function of p62 has been demonstrated to cause increased Abeta42, tau hyperphosphorylation and neurodegeneration in experimental models of Alzheimer’s disease [28]. Notably, an overexpression of p62 has been shown to prevent cognitive decline in transgenic models of AD mice through autophagy-mediated increased neuronal survival and reduced amyloid plaque formation. In addition, p62 was proposed as a multitarget approach for the treatment of AD through the analysis of a recent experimental study with a mouse model of AD [29].

The different forms of frontotemporal dementia are characterized by the aggregation of insoluble proteins within cells. The accumulation of aggregates is often due to genetic mutations in their coding genes. Most inclusions in FTD are characterized by immuno-reactivity, and the neurons and glia of FTD patients are characterized by p62 accumulations.

It has been shown that p62 represents a substrate for macroautophagy; hence, p62 levels increase when macroautophagy is inhibited, and its concentration decreases when macroautophagy is induced [30]. In this regard, the aggregation of p62 may represent a stress response involving selective autophagy [31] and serving a neuroprotective function. In our FTD group, p62 levels were elevated in patients both carrying and not carrying mutations in known causative genes. However, the highest level of p62 was found in one subject carrying a *MAPT* gene mutation. In addition, we found that that CSF p62 levels correlate with biomarkers of neurodegeneration in FTD, and p62 concentrations were positively correlated with t-Tau and p-Tau. Many studies have shown that autophagy deficiency can cause tau accumulation [32,33]. Increasing evidence points to an important role of p62 in the degradation of tau protein by binding polyubiquitinated tau to facilitate its elimination. The spread of tau pathology has been accompanied by the accumulation of p62, and this has been noted in both in vivo and in vitro models [34]. In mouse models, an alteration of p62 has been shown to lead to the accumulation of hyperphosphorylated tau, synaptic deficiencies, and tau tangle-like structures. In addition, increased tau secretion is associated with the accumulation of p62 and LC3, two different markers of autophagic vacuole intermediates [35]. In a recent study, the overexpression of MAPT increased p62 level in primary cultured hippocampal neurons [36]. P62 may be neuroprotective since it promotes autophagic clearance of accumulating tau, shuttling the protein tau to the proteasome for degradation, even if when overexpressed it may be in turn neurotoxic. Thus, an increase in CSF p62 concentration could represent an interesting biomarker for neurodegeneration.

This is the first study to demonstrate an increase in p62 concentrations in the CSF of patients with FTD and AD; in view of the limited samples, the results obtained and their conclusions should be interpreted with caution, and further studies are needed to confirm our data. Autophagy represents a dynamic process, and our results may change according to different stages of neurodegeneration, so studies involving subjects at different stages of disease are desirable. Finally, a further investigation of other different autophagic biomarkers may be crucial for a better understanding of the autophagic machinery in neurodegenerative disorders.

## 5. Conclusions

Our study showed a significant increase of CSF p62 concentrations in patients with Alzheimer’s disease and frontotemporal dementia, supporting an important role of autophagy alterations in these neurodegenerative conditions. In addition, a significant correlation between increased autophagy and clinical characteristics of dementia was found. However, due to the relatively small number of subjects examined, replication studies in larger series are warranted in order to confirm our data.

## Figures and Tables

**Figure 1 brainsci-12-01414-f001:**
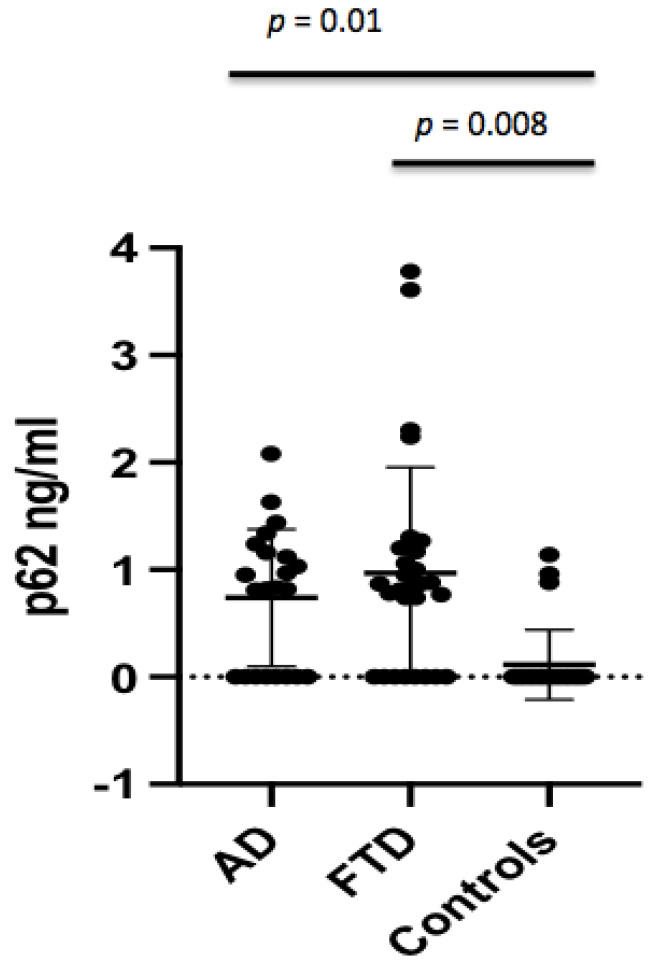
Comparison of CSF p62 concentrations between AD, FTD, and control groups after adjusting for age.

**Figure 2 brainsci-12-01414-f002:**
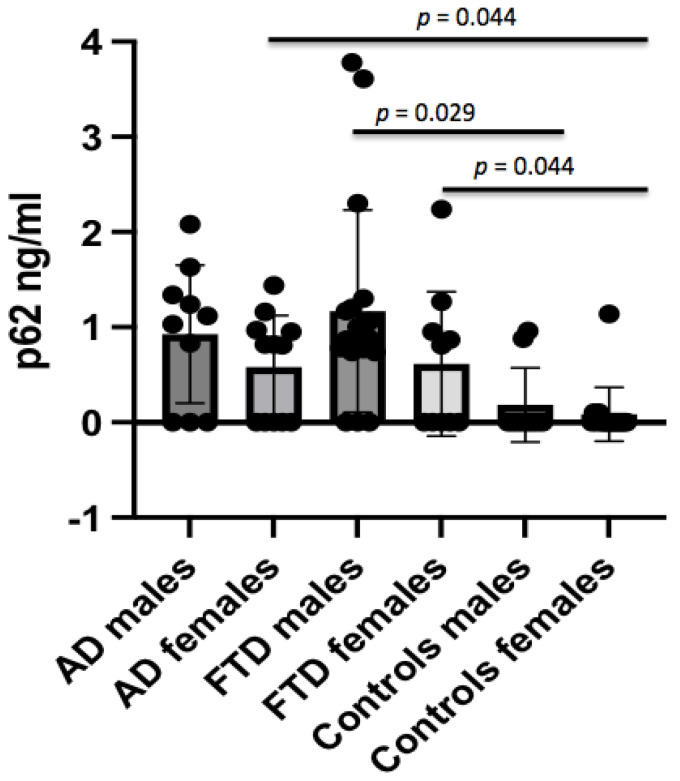
Comparison of p62 concentrations between AD, FTD, and control groups according to gender distribution.

**Figure 3 brainsci-12-01414-f003:**
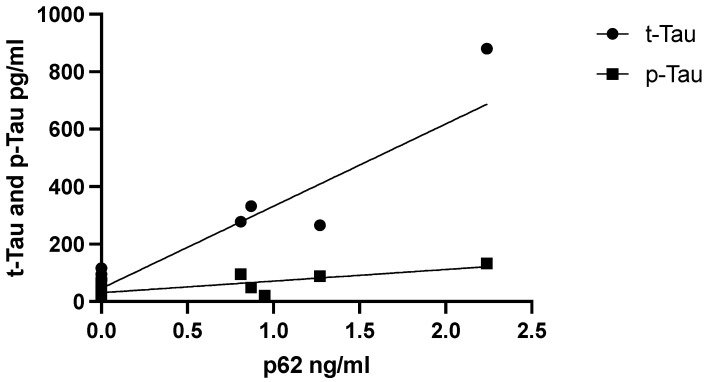
Correlation between CSF p62 and p-Tau and t-Tau concentrations in female FTD patients.

**Figure 4 brainsci-12-01414-f004:**
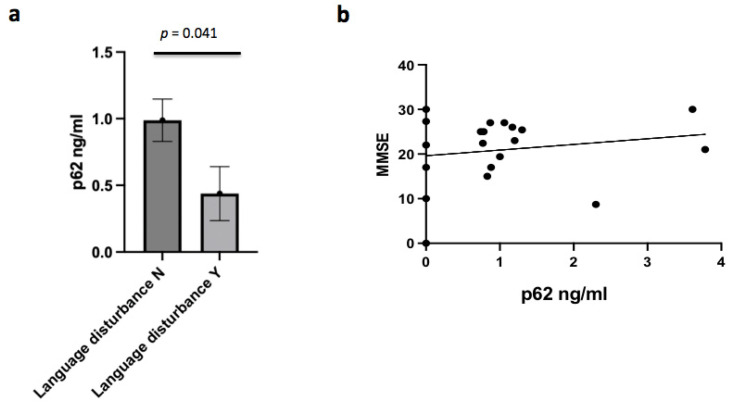
(**a**) Relationship between p62 concentrations and language disturbance (*p* = 0.041). (**b**) Correlation between p62 concentrations and MMSE score in AD patients (NS).

**Table 1 brainsci-12-01414-t001:** Demographic, clinical, and biochemical parameters data of all enrolled subjects.

	AD Patients	FTD Patients	Controls
Age (Years ± SD)	66.4 ± 8.0	64.0 ± 8.0	56.2 ± 15.0 *
Age of onset (Years ± SD)	65.3 ± 8.9	62.4 ± 8.5	-
Sex (Male %)	45	64	38
MMSE (±SD)	19.6 ± 4.5	21.2 ± 6.9	27.1 ± 2.0
CSF parameters			
p62 [ng/mL] (±SD)	0.74 ± 0.64 *	0.97 ± 0.99 *	0.15 ± 0.33
Aβ42 [pg/mL] (±SD)	405.92 ± 94.82 *	977.49 ± 294.61	889.69 ± 282.04
t-Tau [pg/mL] (±SD)	265.86 ± 259.80	156.30 ± 168.29	105.45 ± 35.95
p-Tau [pg/mL] (±SD)	60.70 ± 31.54 *	44.77 ± 26.69	31.81 ± 22.79

SD: standard deviation. AD: Alzheimer’s disease; FTD: frontotemporal dementia; MMSE: Mini Mental State Examination. CSF: cerebrospinal fluid, * *p* < 0.05.

## Data Availability

The data are not publicly available because they contain information that could compromise research participant privacy/consent. Clinical and biochemical data will be available upon request from any qualified investigator.

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
