# Peer review of "Investigating p62 Concentrations in Cerebrospinal Fluid of Patients with Dementia: A Potential Autophagy Biomarker In Vivo?"

_brainsci, 2022, doi:10.3390/brainsci12101414_

Round 1

Reviewer 1 Report

This study from Rubino et. al investigates the correlation between the autophagy marker p62 and clinical and radiological characteristic of AD and FTD. This study is very preliminary, and it lacks important controls. Please find below a list of major concerns that should be addressed:

1. It is not clear why they used  as control subjects people that suffer from neurological disorder and not healthy controls.

2. Fig.1:  Considering that males and females have different pathology and clinical signs in both AD and FTD, the levels of p62 should be plotted togheter for males and females but also separate.

3.  Fig. 2: The author stated that there is no correlation between p-tau 181. But Tau in both FTD and AD it is phosphorylated at many epitopes. Did the authors look at other p-Tau epitopes? If the answer is no, they cannot state that there is not correlation between p62 and p-TAU until they do not look at others p-Tau epitope.

4. Why did the authors not show any graph from the correlations between p62 and clinical characteristics? At least they should add the ones that show significance.

5. Did they look at other autophagic markers in the CSF? If they did not, why?

6. Did they try a correlation between the levels of p62 and the MMSE scores? This could be useful to link p62 with the severity of the disease.

Reviewer 2 Report

In this report, the authors explored the potential use of cerebrospinal fluid (CSF)p62 as an autophagy biomarker for dementia in vivo. While the authors claimed "To date, CSF p62 concentrations have not been investigated in other neurodegenerative disorders" (Line 64), CSF p62 has already been studied in Parkinson's disease (Youn, 2018, Sci Reports) and the use of CSF p62 has already been explored for traumatic brain injury (Au, 2017, Neurocritical Care). In addition, p62 levels in peripheral blood mononuclear cells have also been examined in Parkinson's disease (Vavilova, 2022, Biomolecules). Nevertheless, study on CSF p62 levels in dementia would provide valuable insights into AD and FTD.

A few questions about the study:

1) While the patients were sequenced for GRN, MAPT and C9orf72 genes, were they also sequenced for p62 mutations, which are also known to cause FTD?

2) The control groups contain neurological cases including multiple sclerosis (MS).  It is well-known that autophagy played an important role in the pathogenesis of MS,  the inclusion of MS patients. as controls might not be appropriate for this study.

3) Since the control group is not composed of healthy individuals, it might be appropriate to give a breakdown of the conditions to ensure that autophagy play a role in these conditions

4) Are there information on the state of disease progression for the AD and FTD patients? It would be interesting to see if there is correlation between CSF p62 concentration and disease progression (early stage vs late stage)

5) The authors stated that p62 concentrations were not different in AD males  in respect to male controls (Line 129) while there are significant different in females. A graph would make the data easier to understand.

6) Since there are also difference in AD female patients and FTD males, is it possible to perform correlation studies on these two groups? (Figure 2)

Round 2

Reviewer 1 Report

All my comments have been addressed by the authors.

Reviewer 2 Report

The reviewer would like to thank the authors for their efforts in making the changes and putting in the additional data. This certainly enrich their story and for that I congratulate them.

Just one minor comment:

1) Line 99: should be 12 pm instead of 12 am ?